# Permutations improve performance in three-dimensional bin packing problem

## Abstract

In recent years, with the advent of deep learning and reinforcement learning, researchers have begun to explore the use of deep reinforcement learning to solve the three-dimensional bin packing problem. However, current innovations in the 3D bin packing problem primarily involve modifications to the network architecture or the incorporation of heuristic rules. Efforts to improve performance from the perspective of function approximation are relatively scarce. As is well known, one of the crucial theoretical foundations of deep learning is the ability of neural networks to approximate many functions. As such, we propose a method based on approximation theory that uses permutations to better approximate policy functions, which we refer to as Permutation Packing. Nonetheless, due to the high memory requirements when the number of permutations is large, we also propose a memory-efficient variation of Permutation Packing, which we call Limited-Memory Permutation Packing. Both methods can be efficiently integrated with existing models. We demonstrate the effectiveness of both Permutation Packing and Limited-Memory Permutation Packing from both theoretical and experimental perspectives. Furthermore, based on our theoretical and experimental results, we find that our methods can effectively improve performance even without retraining the model.

## 1 Introduction

Bin packing problems are typically described in terms of the geometric composition of large objects and small items: the large object is defined as empty and needs to be filled with small items. The primary focus is on improving the layout of items during the packing process to maximize benefits. From an engineering perspective, the goal of the packing process is usually to maximize the utilization of raw materials. Even minor improvements in layout can lead to significant material savings and lower production costs, which is of great significance for large manufacturers. With the development of the economy and society, the bin packing problems encountered in real life are becoming increasingly complex, and people are seeking higher benefits. During the development of the entire industry and other sectors, such as timber production, steel production, glass production, etc., transportation naturally plays an important role. Therefore, more complex bin packing problems have emerged and continue to evolve, such as combining two-dimensional bin packing problems with vehicle routing problems, three-dimensional bin packing problems with constraints such as heavy items not pressing lighter ones, and large items not pressing lighter ones, and their variants.

Deep Learning (DL) has achieved impressive success on a variety of traditional AI tasks. This includes classifying images, generating new images, and playing complex games. A common feature of all these tasks is that they involve objects of very high dimensions. In fact, when expressed in mathematical terms, the image classification problem is the problem of approximating a high-dimensional function defined on image space with a set of discrete values corresponding to the category of each image. The dimensionality of the input space is usually three times the number of pixels in the image, where 3 is the dimensionality of the color space. The image generation problem is the problem of generating samples from a distribution given a set of samples from an unknown high-dimensional distribution. The game of Go problem is about solving a kind of Bellman equation in dynamic programming, because the optimal policy satisfies such an equation. For complex games, the Bellman equation is formulated over a vast space. All of these are achieved by using DL to approximate high-dimensional functions accurately.

In recent years, people have begun to explore the use of Deep Reinforcement Learning (DRL) to solve combinatorial optimization problems. Various algorithms have been proposed to solve the three-dimensional bin packing problem (Hu et al., 2017; Duan et al., 2019; Jiang et al., 2021; Zhang et al., 2021; Li et al., 2022; Zhao et al., 2021b;a). However, the innovations of these methods focus on modifying the network architecture, defining new states, and adding heuristic rules. To our knowledge, thus far, no one has demonstrated the effectiveness of their methods from the standpoint of approximation theory. Therefore, we propose our method grounded in the foundations of approximation theory. In the process of using DRL to solve the three-dimensional bin packing problem, the input to the neural network includes the length, width, and height of the box. Obviously, when the length, width, and height of the box are interchanged, the output of an excellent neural network should remain unchanged. Based on this point, combined with permutations, we propose Permutation Packing. Also, as the number of permutations increases, the memory consumption of Permutation Packing significantly increases. Therefore, we use the Taylor series to improve Permutation Packing and propose Limited-Memory Permutation Packing. Permutation Packing and Limited-Memory Permutation Packing can be easily integrated into existing models, and from the experimental results, the performance of the model is improved even without retraining the model after adding Permutation Packing or Limited-Memory Permutation Packing.

Our contributions are mainly in the following two aspects:

1. Combined with Permutation, we propose Permutation Packing, and we demonstrate the effectiveness of Permutation Packing both theoretically and experimentally.

2. Using Taylor expansion, we provide an effective approximation of the policy function in Permutation Packing. Based on this, we propose Limited-Memory Permutation Packing, which significantly reduces memory consumption. Similarly, we demonstrate the effectiveness of Limited-Memory Permutation Packing both theoretically and experimentally.

## 2 RELATED WORK

Algorithms to solve the three-dimensional bin packing problem can be broadly categorized into four types: exact algorithms, approximation algorithms, metaheuristic algorithms, and learning-based algorithms. Exact algorithms are few and far between, offering the highest quality of solutions but at the slowest speeds, currently capable of handling at most 12 boxes. Approximation algorithms are generally designed based on certain heuristic rules. These algorithms tend to solve problems relatively quickly, but the quality of the solutions largely depends on the effectiveness of the heuristic rules. A good heuristic rule requires the algorithm designer to have an in-depth understanding of the bin packing problem. In some cases, we can also perform some theoretical analysis of approximation algorithms, such as time complexity and worst-case performance. Metaheuristic algorithms, such as simulated annealing, genetic algorithms, particle swarm optimization, typically require a longer time and cannot guarantee the quality of the solution. Their advantage is that they usually improve the solution over time and perform well in the absence of good heuristic rules. Learning-based algorithms, generally speaking, are faster than metaheuristic algorithms and can even be faster than most approximation algorithms, but the model training time is long. When the length, width, or height of the bin change, or when the distribution of the box's length, width, or height varies, the model must be retrained. Furthermore, due to the large action space, learning-based algorithms can currently handle at most 120 boxes during training because of the complexity of the 3D BPP. Recently proposed algorithms are basically hybrid algorithms and cannot be classified as a single type among the four types of algorithms, but combine the advantages of two or three types of algorithms.

Due to space constraints, we will only introduce some representative papers for each type of algorithm. The current most accurate algorithm (Silva et al., 2019) takes several hours to solve a packing problem with only 12 items, which is obviously not efficient enough for practical use. For approximation algorithms, Parreño et al. (2008) proposed Empty Maximal Space. Empty Maximal Space strictly requires boxes to be placed only in certain specified positions. At the time, Empty Maximal Space significantly improved packing efficiency. Even now, many algorithms only choose from a few positions specified by Empty Maximal Space when deciding where to place boxes. There are the most metaheuristic algorithms, with common metaheuristic algorithms being used, such as simulated annealing (Fenrich et al., 1989; Zhang et al., 2007), genetic algorithms (Kang et al., 2012; Corcoran III & Wainwright, 1992; Whitley & Starkweather, 1990; Karabulut & İnceoğlu, 2004;

Gonçalves & Resende, 2013; Wu et al., 2010; de Andoin et al., 2022), ant colony optimization Silveira et al. (2013), and quantum algorithms (De Andoin et al., 2022; Bozhedarov et al., 2023; V. Romero et al., 2023). As for learning-based algorithms, Hu et al. (2017) was the first to use deep reinforcement learning to solve the 3D BPP. The work of (Duan et al., 2019; 2022; Zhang et al., 2021; Jiang et al., 2021; Zhao et al., 2021b;a) is based on (Hu et al., 2017), modifying the network structure, redefining the state in the Markov process, and adding some heuristic rules.

## 3 BACKGROUND

### 3.1 APPROXIMATION THEORY

Compared to $5$, $4$ is closer to $3$ because $|4 - 3| < |5 - 3|$. For functions $f(\boldsymbol{x})$, $g(\boldsymbol{x})$, and $h(\boldsymbol{x})$, assuming their domain is $\Omega \in \mathbb{R}^m$ and their images are one-dimensional, we want to know which of $f$ and $g$ is closer to $h$ in $\Omega$.

First, we define $||f(\boldsymbol{x})||_{\Omega,p} = \left( \int_\Omega |f(\boldsymbol{x})|^p d\boldsymbol{x} \right)^{1/p}$. To evaluate whether $f$ or $g$ is closer to $h$, we merely need to compare the magnitudes of $||f - h||_{\Omega,p}$ and $||g - h||_{\Omega,p}$.

The values of $p$ we commonly use are 1, 2, and $+\infty$. When $p = +\infty$, we have $||f||_{\Omega,p} = \sup_{\boldsymbol{x} \in \Omega} |f(\boldsymbol{x})|$.

### 3.2 DEFINITION OF THE PROBLEM

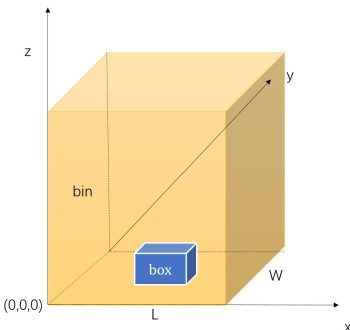

Figure 1: The coordinate system of bin.

All rectangular cuboid boxes are required to be packed into a rectangular cuboid bin. The length and width of the bin are predetermined, but the height is flexible. The length and width of the bin are denoted by $L$ and $W$ respectively. A coordinate system is employed to describe the positions of the items and the bin, with $(0, 0, 0)$ defined as the coordinates of the left-front-bottom vertex of the bin. There are $N$ boxes, with the length, width, and height of the $i$-th box denoted as $l_i, w_i, h_i$ respectively. Once all boxes are packed into the bin, the coordinates of each item are fixed. $(x_i, y_i, z_i)$ represents the coordinates of the right-rear-top vertex of the $i$-th item. Figure 1 visually illustrates the coordinate axes. The objective is to:

$$\min H \tag{1}$$

where $H = \max_{i \in \{1,2,...,N\}} z_i$.

The constraints include:

1. Boxes must be entirely within the bin.

2. Boxes must not overlap each other.

3. Diagonal placement of boxes is not permitted.

If the length, width, and height of all boxes are known from the outset, the problem is classified as an offline 3D BPP. Conversely, if the dimensions of just one box are known at each decision stage, indicating a fixed packing sequence, the problem is recognized as an online 3D BPP.

### 3.3 FORMULATING 3D BPP AS A MARKOV DECISION PROCESS

**State:** In the 3D BPP, the state can be divided into two categories: the state of the boxes that have not yet been packed into the bin, usually described by their length, width, and height; and the state of the boxes that have been packed into the bin. For the state of boxes that have been loaded into the bin, in addition to being described by their length, width, and height, it also involves the internal conditions of the bin. There are two main ways to describe the interior of the bin currently, one is using a height map (Zhao et al., 2021a), which can be seen as an overhead view of the bin. Using the concept of pixels in an image, each pixel in the height map represents a height; the other is to directly give the position and orientation of each box in the bin (Li et al., 2022). We denote the state at time $t$ as $\mathbf{S}_t$.

**Action:** In the offline 3D BPP, actions can be divided into three categories. The first is index action. We select one from $N$ boxes; the second is orientation action. As shown in Figure 2, there are 6 orientation actions; the third is position action. Position action can be regarded as choosing the $x$ and $y$ coordinates of the bottom left vertex of the box. As for why the $z$ coordinate is not involved, if the same $x$ and $y$ coordinates correspond to multiple $z$ coordinates, the smallest $z$ coordinate is chosen by default. In the online 3D BPP, since the packing order is fixed, there is no index action, only orientation action and position action. We denote the action at time $t$ as $a_t$. The decision on which action to take is determined by the policy function $\pi(\mathbf{S}_t)$, which is approximated by a neural network. The output of the neural network is a probability vector, each component of which is non-negative, and the sum of all components is 1. Each component represents the probability of taking the corresponding action.

**Reward:** The reward is set as the loading rate $r_u = \frac{\sum_{i=1}^{N} l_i w_i h_i}{LWH}$.

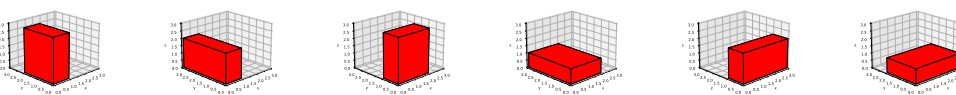

Figure 2: Six orientations.

## 4 METHOD

Firstly, we provide the definition of permutation.

**Definition 1 (permutation)** *A permutation of a set S is defined as a bijection from S to itself.*

We assume $S = \{1, 2, ..., n\}$ and consider a permutation $\sigma$ of $S$, where $\sigma(i) = j_i$. We define the action of a permutation on a vector such that if $\boldsymbol{x} = (x_1, x_2, ..., x_n) \in \mathbb{R}^n$, then $\sigma(\boldsymbol{x}) = (x_{j_1}, x_{j_2}, ..., x_{j_n})$. Following this, we define a fixed permutation.

**Definition 2 (fixed permutation)** *$f$ is a function, $\Omega \subseteq \mathbb{R}^n$, $\boldsymbol{x} \in \Omega$ and $f(\boldsymbol{x}) \in \mathbb{R}$. A permutation $\sigma$ is said to be fixed permutation of $f$ in $\Omega$ if $f(\sigma(\boldsymbol{x})) = f(\boldsymbol{x}), \forall \boldsymbol{x} \in \Omega$.*

It is clear that the identity mapping belongs to the category of fixed permutations. Subsequently, we present the following theorem:

**Theorem 1** *$f^*, f$ is a function, $\Omega \subseteq \mathbb{R}^n$, $\boldsymbol{x} \in \Omega$, $f(\boldsymbol{x}) \in \mathbb{R}$ and $p \geq 1$. $\sigma_1, \sigma_2, ..., \sigma_m$ is fixed permutation of $f^*$ in $\Omega$. $\forall i \in \{1, 2..., m\}, \sigma_i(\Omega) = \Omega$. $f, f^*, \Omega$ is measurable. $||f||_{\Omega,p} < +\infty$ and $||f^*||_{\Omega,p} < +\infty$. Then we have*

$$||\frac{\sum_{i=1}^{m} f(\sigma_i(\boldsymbol{x}))}{m} - f^*(\boldsymbol{x})||_{\Omega,p} \leq ||f(\boldsymbol{x}) - f^*(\boldsymbol{x})||_{\Omega,p} \qquad (2)$$

*If $p > 1$ and measure of $\Omega$ is larger than 0 ,Equality holds if and only if $f(\sigma_1(\boldsymbol{x}) - f^*(\sigma_1(\boldsymbol{x})), f(\sigma_2(\boldsymbol{x})) - f^*(\sigma_2(\boldsymbol{x})), ..., f(\sigma_m(\boldsymbol{x})) - f^*(\sigma_m(\boldsymbol{x}))$ are proportional.*

In Theorem 1, $f$ can be viewed as a neural network and $f^*$ is considered as the ideal policy function. Theoretically, when the length, width, and height of a box are interchanged, the output of the ideal policy function should remain the same since the box itself does not change. The input to a neural network can be considered as a vector encompassing the length, width, and height of all boxes. The swapping of the dimensions – length, width, and height – can be viewed as a form of *permutation*. After applying this permutation to the input, the output remains unchanged. This implies that the ideal policy function possesses *fixed permutations* that are not identity mapping.

As for the conditions under which equality holds in Theorem 1, given that $f$ is a neural network, it is virtually impossible to meet the condition that $f(\sigma_1(\boldsymbol{x}) - f^*(\sigma_1(\boldsymbol{x})), f(\sigma_2(\boldsymbol{x})) - f^*(\sigma_2(\boldsymbol{x})), ..., f(\sigma_m(\boldsymbol{x})) - f^*(\sigma_m(\boldsymbol{x}))$ are proportional, as long as these $m$ fixed permutations are not all identical. Hence, using $\frac{\sum_{i=1}^m f(\sigma_i(\boldsymbol{x}))}{m}$ to approximate the ideal policy function is more effective.

Based on this, we designed Permutation Packing, as shown in Algorithm 1. In Algorithm 1, RandomInstance() denotes the random generation of a sample. SampleRollout($\pi_{t,i}$) is indicative of an action being randomly chosen based on $\pi_{t,i}$, with the probability of each action being chosen equal to the numerical value of its corresponding component. $\pi_{t,i}[a_{t,i}]$ represents the $a_{t,i}$-th component of $\pi_{t,i}$. The term $r_u(\mathbf{S}_{N+1,i})$ represents the loading rate when the state is $r_u(\mathbf{S}_{N+1,i})$, while $b(\mathbf{S}_{1,i})$ denotes the baseline for the sample $\mathbf{S}_{1,i}$.

---

**Algorithm 1** Permutation Packing

---

**Input:** neural network $f(\mathbf{S}; \boldsymbol{\theta})$, batch size $B$, number of Epochs $E$, significance $\alpha$, the quantity of boxes $N$, fixed permutation $\sigma_1, \sigma_2, ..., \sigma_m$
1: Init $\boldsymbol{\theta}$
2: **for** epoch = $1, 2, .., E$ **do**
3:     $\mathbf{S}_{1,i} \leftarrow$ RandomInstance(), $\forall i \in \{1, 2, ..., B\}$
4:     **for** t = $1, .., N$ **do**
5:         $\pi_{t,i} \leftarrow \frac{\sum_{j=1}^m f(\sigma_j(\mathbf{S}_{t,i}), \boldsymbol{\theta})}{m}, \forall i \in \{1, 2, ..., B\}$
6:         $a_{t,i} \leftarrow$ SampleRollout($\pi_{t,i}$), $\forall i \in \{1, 2, ..., B\}$
7:         According to $a_{t,i}$, Update $\mathbf{S}_{t+1,i}$;
8:     **end for**
9:     $\nabla L \leftarrow \frac{1}{B} \sum_{i=1}^B \left( b(\mathbf{S}_{1,i}) - r_u(\mathbf{S}_{N+1,i}) \right) \nabla_{\boldsymbol{\theta}} \left( \sum_{t=1}^N \log \pi_{t,i}[a_{t,i}] \right)$
10:     $\boldsymbol{\theta} \leftarrow$ Adam($\boldsymbol{\theta}, \nabla L$)
11: **end for**

---

We observe that in comparison to the original algorithm, which requires only one input, Permutation Packing involves $m$ inputs. When $m$ is large, this significantly increases memory consumption. Hence, with a view to reducing memory usage, we aim to preserve the single-input characteristic of the original algorithm.

**Theorem 2** *$f$ is a function $\Omega \subseteq \mathbb{R}^n$, $\boldsymbol{x}_i \in \Omega$, $i = 1, 2, ..., m$. $\boldsymbol{x}_0 = \frac{\sum_{i=1}^m \boldsymbol{x}_i}{m}$ $f$ is 3 times differentiable at $\boldsymbol{x}_0$.*

$$\frac{\sum_{i=1}^m f(\boldsymbol{x}_i)}{m} = f(\boldsymbol{x}_0) + \frac{1}{2m} \sum_{i=1}^m (\boldsymbol{x}_i - \boldsymbol{x}_0)^T H_f(\boldsymbol{x}_0) \boldsymbol{x}_i + o(\max_{i \in \{1, 2, ..., m\}} ||x_i - x||^3) \quad (3)$$

*where $H_f(\boldsymbol{x}_0)$ denotes the Hessian matrix of function $f$ at point $\boldsymbol{x}_0$.*

According to Theorem 2, we can approximate $\frac{\sum_{i=1}^m f(\sigma_i(\boldsymbol{x}))}{m}$ with $f(\frac{\sum_{i=1}^m \sigma_i(\boldsymbol{x})}{m}) + \frac{1}{2m} \sum_{i=1}^m (\sigma_i(\boldsymbol{x}) - \frac{\sum_{i=1}^m \sigma_i(\boldsymbol{x})}{m})^T H_f(\frac{\sum_{i=1}^m \sigma_i(\boldsymbol{x})}{m}) \sigma_i(\boldsymbol{x})$.

However, in Permutation Packing, the last layer of the neural network $f(\mathbf{S}; \boldsymbol{\theta})$ is a Softmax function, meaning each component of $f(\sigma_j(\mathbf{S}_{t,i}), \boldsymbol{\theta})$ ,$j = 1, 2, ..., m$ is non-negative and the sum of all

components equals 1. Therefore, for $\pi_{t,i} = \frac{\sum_{j=1}^{m} f(\sigma_j(\mathbf{S}_{t,i}), \boldsymbol{\theta})}{m}$, each component of $\pi_{t,i}$ is non-negative and the sum of all components equals 1.

For $f(\frac{\sum_{i=1}^{m} \sigma_i(\boldsymbol{x})}{m}) + \frac{1}{2m} \sum_{i=1}^{m} (\sigma_i(\boldsymbol{x}) - \frac{\sum_{i=1}^{m} \sigma_i(\boldsymbol{x})}{m})^T H_f(\frac{\sum_{i=1}^{m} \sigma_i(\boldsymbol{x})}{m}) \sigma_i(\boldsymbol{x})$, it's challenging to ensure that each component is non-negative and the sum equals 1. As a result, we consider the neural network without the last Softmax layer and introduce the Limited-Memory Permutation Packing, as shown in Algorithm 2. Computing the Hessian Matrix is challenging in common deep learning frameworks. For $H_g(\frac{\sum_{j=1}^{m} \sigma_j(\mathbf{S}_{t,i})}{m}, \boldsymbol{\theta})$ in Algorithm 2, we can employ Equation 4 to avoid direct computation of the Hessian Matrix.

$$x^T H_f(x_0) y = \nabla \left( x^T \nabla f(x_0) \right) y \tag{4}$$

In Equation 4, $x, y$ are vectors composed of constants, and $\nabla f(x_0)$ denotes the gradient of function $f$ at $x_0$. By applying Equation 4, we can achieve our objective by simply calculating the gradient twice.

---

**Algorithm 2** Limited-Memory Permutation Packing

---

**Input:** neural network without last Softmax layer $g(\mathbf{S}; \boldsymbol{\theta})$, batch size $B$, number of Epochs $E$, steps per epoch $T$, significance $\alpha$, the quantity of boxes $N$, fixed permutation $\sigma_1, \sigma_2, ..., \sigma_m$
1: Init $\boldsymbol{\theta}$
2: **for** epoch = 1, 2, .., $E$ **do**
3:     $\mathbf{S}_{1,i} \leftarrow$ RandomInstance()
4:     **for** t = 1, .., $N$ **do**
5:         $\pi_{t,i} \leftarrow Softmax\big(g(\frac{\sum_{j=1}^{m} \sigma_j(\mathbf{S}_{t,i})}{m}, \boldsymbol{\theta}) + \frac{1}{2m} \sum_{j=1}^{m} (\sigma_j(\mathbf{S}_{t,i}) - \frac{\sum_{j=1}^{m} \sigma_j(\mathbf{S}_{t,i})}{m})^T H_g(\frac{\sum_{j=1}^{m} \sigma_j(\mathbf{S}_{t,i})}{m}, \boldsymbol{\theta}) \sigma_j(\mathbf{S}_{t,i})\big), \forall i \in \{1, 2, ..., B\}$
6:         $a_{t,i} \leftarrow$ SampleRollout$(\pi_{t,i})$
7:         According to $a_{t,i}$, Update $\mathbf{S}_{t+1,i}$
8:     **end for**
9:     $\nabla L \leftarrow \frac{1}{B} \sum_{i=1}^{B} \left( b(\mathbf{S}_{1,i}) - r_u(\mathbf{S}_{N+1,i}) \right) \nabla_{\boldsymbol{\theta}} (\sum_{t=1}^{N} \log \pi_{t,i}[a_{t,i}])$
10:     $\boldsymbol{\theta} \leftarrow$ Adam$(\boldsymbol{\theta}, \nabla L)$
11: **end for**

---

# 5 EXPERIMENT

## 5.1 EXPERIMENTAL PARAMETER SETTING

We instantiated our approach on two state-of-the-art network architectures, namely RCQL (Li et al., 2022) and attend2pack (Zhang et al., 2021).

Both RCQL and attend2pack utilized Adam (Kingma & Ba, 2014) as the optimizer, and the training was guided by the Rollout algorithm (Kool et al., 2019). When the $p$ value in the Rollout algorithm exceeded 0.95, the learning rate was reduced by 5%. The Rollout algorithm was executed for 500 epochs, each consisting of 10 steps, a batch size of 64, and a significance level of 0.05. Both models employed LayerNorm (Ba et al., 2016) for normalization. They incorporated multi-head attention layers or their variants, with 8 heads, each of size 16, and 3 such layers or variants. The feedforward layer consisted of two fully connected layers, with output dimensions of 512 and 128, and activation functions being ReQUr (Yu et al., 2021) and ReQU (Li et al., 2019). The test set encompassed 16384 samples, with the lengths, widths, and heights of the boxes in both the test and training sets randomly generated with equal probability from integers between 10 and 50.

For attend2pack-specific hyperparameters, the initial learning rate was set at $10^{-5}$. The $C$ value from the original paper (Zhang et al., 2021) was 10. The model included 3 convolutional layers, all with 4 output channels. The first convolutional layer had 2 input channels, while the second and third convolutional layers had 4 input channels.

For RCQL-specific hyperparameters, the initial learning rate was $10^{-4}$, the length of the recurrent FIFO queue was 20, the context size of the packed state was 30, and the context size of the unpacked state was $\min N, 100$.

For $\mathbf{S}_t$, we have:

$$\mathbf{S}_t = (l_1, l_2, ..., l_N, w_1, w_2, ..., w_N, h_1, h_2, ..., h_N, ...)$$

We used a maximum of six fixed permutations, where $\sigma_1$ is the identity mapping, and the other five fixed permutations are as follows:

$$\sigma_2(\mathbf{S}_t) = (l_1, l_2, ..., l_N, h_1, h_2, ..., h_N, w_1, w_2, ..., w_N, ...)$$
$$\sigma_3(\mathbf{S}_t) = (w_1, w_2, ..., w_N, l_1, l_2, ..., l_N, h_1, h_2, ..., h_N, ...)$$
$$\sigma_4(\mathbf{S}_t) = (w_1, w_2, ..., w_N, h_1, h_2, ..., h_N, l_1, l_2, ..., l_N, ...)$$
$$\sigma_5(\mathbf{S}_t) = (h_1, h_2, ..., h_N, l_1, l_2, ..., l_N, w_1, w_2, ..., w_N, ...)$$
$$\sigma_6(\mathbf{S}_t) = (h_1, h_2, ..., h_N, w_1, w_2, ..., w_N, l_1, l_2, ..., l_N, ...)$$

In Tables 1 and 2, we only used $\sigma_1$ and $\sigma_2$. In Table 3, when the number of fixed permutations is $m$, we used $\sigma_1, \sigma_2, ..., \sigma_m$.

Our comparison encompassed both traditional and learning-based algorithms: 1) Genetic Algorithm with Deepest Bottom Left Heuristic (GA+DBLF) (Wu et al., 2010), where the population size and number of generations were set at 120 and 200 respectively; 2) Extreme Point (EP) (Crainic et al., 2008); 3) Largest Area Fit First (LAFF) (Gürbüz et al., 2009); 4) EB-AFIT packing algorithm (Baltacioglu, 2001); 5) MTSL (Duan et al., 2019); 6) Multimodal (MM) (Jiang et al., 2021); 7) attend2pack (A) (Zhang et al., 2021); 8) RCQL (R) (Li et al., 2022).

## 5.2 EXPERIMENTAL RESULTS

Table 1: The experimental outcomes for varying $N$ with $L = 120, W = 100$. Both figures outside and inside parentheses exclude the % symbol. The figure outside the parentheses denotes the average loading rate following seven experimental repetitions, while the figure within the parentheses signifies the standard deviation.

| $N$ | 25 | 50 | 100 |
|---|---|---|---|
| GA+DBLF | 60.24(1.74) | 62.43(1.61) | 65.87(1.76) |
| EP | 62.64(0) | 61.69(0) | 62.87(0) |
| LAFF | 62.48(0) | 61.40(0) | 60.67(0) |
| EB-AFIT | 61.36(0) | 60.71(0) | 63.06(0) |
| MTSL | 65.48(2.04) | 65.71(2.82) | 51.19(2.39) |
| MM | 68.00(2.08) | 68.32(2.60) | 68.65(1.58) |
| A | 72.96(1.38) | 74.29(1.55) | 73.40(3.49) |
| A+P(Our) | 74.15(2.45) | 74.98(2.23) | 74.38(2.52) |
| A+P+T(Our) | **76.59**(2.38) | **76.51**(1.53) | **76.76**(2.03) |
| A+LP(Our) | 74.17(2.09) | 74.65(2.17) | 74.22(1.67) |
| A+LP+T(Our) | 76.46(2.31) | 76.09(1.45) | 76.39(2.05) |
| R | 69.76(2.57) | 70.37(1.78) | 71.06(3.81) |
| R+P(Our) | 71.57(1.05) | 71.59(2.28) | 72.77(1.77) |
| R+P+T(Our) | 73.46(1.22) | 73.84(2.98) | 74.00(2.56) |
| R+LP(Our) | 71.24(2.98) | 71.24(1.78) | 72.40(2.20) |
| R+LP+T(Our) | 73.28(1.96) | 73.58(2.33) | 73.66(1.78) |

Table 2: The experimental outcomes for varying $L, W$ with $N = 100$. The numerical representations in the table retain the same meanings as those outlined in Table 1.

| (L,W) | (140,120) | (160,140) | (180,160) |
|---|---|---|---|
| GA+DBLF | 60.14(1.96) | 63.34(3.78) | 61.69(2.29) |
| EP | 62.93(0) | 65.18(0) | 64.22(0) |
| LAFF | 61.42(0) | 62.46(0) | 62.15(0) |
| EB-AFIT | 62.49(0) | 60.39(0) | 62.66(0) |
| MTSL | 59.97(3.89) | 61.13(2.27) | 55.18(1.84) |
| MM | 70.74(2.90) | 69.06(3.17) | 70.28(3.35) |
| A | 73.88(2.02) | 74.92(2.60) | 74.79(2.30) |
| A+P(Our) | 75.86(1.71) | 75.18(2.24) | 75.61(1.49) |
| A+P+T(Our) | **77.56**(1.03) | **77.36**(1.52) | **77.60**(1.39) |
| A+LP(Our) | 75.31(2.82) | 74.51(2.29) | 75.62(2.85) |
| A+LP+T(Our) | 77.13(2.01) | 77.22(2.69) | 77.57(2.26) |
| R | 72.61(1.70) | 71.78(2.67) | 72.65(2.31) |
| R+P(Our) | 73.37(2.86) | 73.16(2.96) | 73.49(1.63) |
| R+P+T(Our) | 75.52(1.99) | 75.58(2.39) | 76.06(1.23) |
| R+LP(Our) | 73.29(1.18) | 72.88(1.99) | 73.03(2.58) |
| R+LP+T(Our) | 75.15(2.13) | 75.18(2.90) | 75.48(2.89) |

Tables 1 and 2 present the results of our experiments. In our methodology, "P" signifies Permutation Packing, while "LP" stands for Limited-Memory Permutation Packing. The absence of "T" implies that our method is only used during the testing stage, while the presence of "T" indicates that our method is employed during both the training and testing stages. From table, it is evident that the strategy "A+LP+T" delivers the best performance across various settings of "N", "L" and "W". Comparing Permutation Packing and Limited-Memory Permutation Packing, we find that the performance of Limited-Memory Permutation Packing is not as good. This is due to Limited-Memory Permutation Packing being an approximation of Permutation Packing. Next, when we compare the effects of incorporating our method solely during the testing stage versus both stages, it is clear that using our method during both stages outperforms using it solely in the testing stage. Finally, comparing the effects of integrating our method only during the testing stage with the original model shows that even when our method is applied only during testing, performance is enhanced. Hence, even if one does not wish to retrain the model, using our method can still boost its performance.

Table 3 presents the experimental results for varying numbers of fixed permutations. As can be observed, increasing the number of fixed permutations can enhance the model's performance, albeit the magnitude of improvement is not substantial.

Table 3: The experimental outcomes for varying $m$ with $L = 120, W = 100, N = 100$. Both figures outside and inside parentheses exclude the % symbol. The figure outside the parentheses denotes the average loading rate following seven experimental repetitions, while the figure within the parentheses signifies the standard deviation.

| $m$ | 2 | 3 | 4 | 5 | 6 |
|---|---|---|---|---|---|
| A+P | 74.38(2.52) | 74.71(1.41) | 74.82(1.20) | 74.94(2.92) | 75.08(2.94) |
| A+P+T | 76.76(2.03) | 76.87(2.18) | 76.88(2.25) | 76.96(1.10) | 77.07(1.43) |
| A+LP | 74.22(1.67) | 74.67(2.11) | 74.81(2.87) | 74.86(2.56) | 74.94(2.29) |
| A+LP+T | 76.39(2.05) | 76.63(1.20) | 76.83(2.17) | 76.96(1.93) | 77.00(2.67) |

Table 4 provides the memory usage during training for different numbers of fixed permutations. It is evident that as $m$ increases, the memory consumption of Limited-Memory Permutation Packing is significantly less than that of Permutation Packing.

Table 4: The details of memory consumption during model training when the batch size is set to 32 and $N = 100$. Under these conditions, the memory usage of the original model stands at 1.4G.

| $m$ | 2 | 3 | 4 | 5 | 6 |
|---|---|---|---|---|---|
| A+P+T | 2.1G | 3.1G | 4.0G | 5.0G | 5.9G |
| A+LP+T | 1.4G | 1.6G | 1.8G | 1.9G | 2.1G |

Table 5 presents the per-step training time for a batch size of 64, with $L = 120$, $W = 100$, and $N = 100$. It can be observed that the training time of Permutation Packing aligns closely with that of the original model, while the training time for the Limited-Memory Permutation Packing is approximately 7% longer than the original model.

Table 5: The per-step training time for a batch size of 64, with $L = 120$, $W = 100$, and $N = 100$.

| method | A | A+P+T | A+LP+T | R | R+P+T | R+LP+T |
|---|---|---|---|---|---|---|
| time(s) | 510 | 518 | 542 | 428 | 441 | 462 |

## 6 CONCLUSION

In this paper, we propose Permutation Packing and Limited-Memory Permutation Packing based on approximation theory. Both methods can be seamlessly integrated with existing models to improve performance. Remarkably, considerable performance enhancement is still achieved even without retraining the models. Although the performance of Limited-Memory Permutation Packing is less than that of Permutation Packing, its memory consumption is significantly reduced when the number of permutations is high.

Theoretically speaking, our method is not only applicable to three-dimensional bin packing problems but can also be extended to other tasks. For instance, it can be seamlessly extended to two-dimensional bin packing problems, where the ideal strategy function's output should remain invariant when the length and width inputs to the network are swapped in the two-dimensional bin packing problem. It also has the potential for straightforward extension to other domains, in image classification problems, image data is typically square. For most datasets, images rotated by $\frac{\pi}{2}, \pi, \frac{3\pi}{2}$ should belong to the same class, and these rotations can be viewed as permutations. In addressing partial differential equation problems using neural networks, we can identify fixed permutations of non-identity mapping in some partial differential equations. For example, in the equation $\Delta u = 0$, $(x, y) \in \Omega = [0, 1] \times [0, 1]$ with boundary conditions $u(x, y) = f(x, y)$, $(x, y) \in \partial\Omega$, where $f(x, y)$ is known and $f(x, y) = f(y, x)$, we can infer that $u(x, y) = u(y, x)$.

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

# A   APPENDIX

## A.1   PROOF OF THEOREM 1

**Proof** *Let $h(\boldsymbol{x}) = f(\boldsymbol{x}) - f^*(\boldsymbol{x})$, $\forall i \in \{1, 2, ..., m\}$*

$$
\begin{aligned}
||h(\boldsymbol{x})||_{\Omega,p} &= \left( \int_\Omega |h(\boldsymbol{x})|^p d\boldsymbol{x} \right)^{\frac{1}{p}} \\
&= \left( \int_\Omega |h(\sigma_i(\boldsymbol{x}))|^p d\sigma_i(\boldsymbol{x}) \right)^{\frac{1}{p}} \\
&= \left( \int_{\sigma_i^{-1}(\Omega)} |h(\sigma_i(\boldsymbol{x}))|^p \left| det(\frac{\partial \sigma_i(\boldsymbol{x})}{\partial \boldsymbol{x}}) \right| d\boldsymbol{x} \right)^{\frac{1}{p}} \\
&= \left( \int_\Omega |h(\sigma_i(\boldsymbol{x}))|^p d\boldsymbol{x} \right)^{\frac{1}{p}} = ||h(\sigma_i(\boldsymbol{x}))||_{\Omega,p}
\end{aligned}
\tag{5}
$$

*In Equation 5, the fourth equality sign is justified by the properties of the Jacobian matrix $\frac{\partial \sigma_i(\boldsymbol{x})}{\partial \boldsymbol{x}}$. Given the definition of permutation, each row and column of this matrix contains only one element with the value 1, while all other elements are 0. According to the definition of the determinant, the determinant of $\frac{\partial \sigma_i(\boldsymbol{x})}{\partial \boldsymbol{x}}$ is either 1 or -1. Thus, the absolute value of the determinant of $\frac{\partial \sigma_i(\boldsymbol{x})}{\partial \boldsymbol{x}}$ equals 1. Considering that $\sigma_i$ is a bijection and $\sigma_i(\Omega) = \Omega$, it can be shown that $\Omega = \sigma_i^{-1}(\Omega)$.*

$$
\begin{aligned}
||\frac{\sum_{i=1}^m f(\sigma_i(\boldsymbol{x}))}{m} &- f^*(\boldsymbol{x})||_{\Omega,p} \\
&= \left( \int_\Omega \left| \frac{\sum_{i=1}^m f(\sigma_i(\boldsymbol{x}))}{m} - f^*(\boldsymbol{x}) \right|^p d\boldsymbol{x} \right)^{\frac{1}{p}} \\
&= \left( \int_\Omega \left| \frac{\sum_{i=1}^m f(\sigma_i(\boldsymbol{x}))}{m} - \frac{m f^*(\boldsymbol{x})}{m} \right|^p d\boldsymbol{x} \right)^{\frac{1}{p}} \\
&= \left( \int_\Omega \left| \frac{\sum_{i=1}^m f(\sigma_i(\boldsymbol{x}))}{m} - \frac{\sum_{i=1}^m f^*(\sigma_i(\boldsymbol{x}))}{m} \right|^p d\boldsymbol{x} \right)^{\frac{1}{p}} \\
&= \left( \int_\Omega \left| \frac{\sum_{i=1}^m h(\sigma_i(\boldsymbol{x}))}{m} \right|^p d\boldsymbol{x} \right)^{\frac{1}{p}} \\
&= \frac{1}{m} \left( \int_\Omega \left| \sum_{i=1}^m h(\sigma_i(\boldsymbol{x})) \right|^p d\boldsymbol{x} \right)^{\frac{1}{p}} \\
&= \frac{1}{m} || \sum_{i=1}^m h(\sigma_i(\boldsymbol{x}))||_{\Omega,p} \\
&\leq \frac{1}{m} \left( \sum_{i=1}^m ||h(\sigma_i(\boldsymbol{x}))||_{\Omega,p} \right) \\
&= ||h(\boldsymbol{x})||_{\Omega,p} = ||f(\boldsymbol{x}) - f^*(\boldsymbol{x})||_{\Omega,p}
\end{aligned}
\tag{6}
$$

*In Equation 6, the " $\leq$ " is a manifestation of the Minkowski inequality. Within the context of the Minkowski inequality, for $p > 1$ and the measure of $\Omega$ greater than zero, equality is achieved if and only if $h(\sigma_1(\boldsymbol{x})), h(\sigma_2(\boldsymbol{x})), ..., h(\sigma_m(\boldsymbol{x}))$ are all proportional to each other. The penultimate equality arises from the application of Equation 5.* $\square$

## A.2   PROOF OF THEOREM 2

**Proof** *Firstly, we conduct a Taylor expansion of function $f$ at point $\boldsymbol{x}_0$:*

$$
f(\boldsymbol{x}) = f(\boldsymbol{x}_0) + \nabla f(\boldsymbol{x}_0)^T (\boldsymbol{x} - \boldsymbol{x}_0) + \frac{1}{2}(\boldsymbol{x} - \boldsymbol{x}_0)^T H_f(\boldsymbol{x}_0)(\boldsymbol{x} - \boldsymbol{x}_0) + o(||\boldsymbol{x} - \boldsymbol{x}_0||^3)
\tag{7}
$$

*Here, $\nabla f(\boldsymbol{x}_0)$ denotes the gradient of $f$ at $\boldsymbol{x}_0$. Thus, we obtain:*

$$\sum_{i=1}^{m} f(\boldsymbol{x}_m) = \sum_{i=1}^{m} \left( f(\boldsymbol{x}_0) + \nabla f(\boldsymbol{x}_0)^T (\boldsymbol{x}_i - \boldsymbol{x}_0) + \frac{1}{2} (\boldsymbol{x}_i - \boldsymbol{x}_0)^T H_f(\boldsymbol{x}_0)(\boldsymbol{x}_i - \boldsymbol{x}_0) \right.$$
$$\left. + o(\|\boldsymbol{x}_i - \boldsymbol{x}_0\|^3) \right)$$
$$= m f(\boldsymbol{x}_0) + \nabla f(\boldsymbol{x}_0)^T (\sum_{i=1}^{m} \boldsymbol{x}_i - m\boldsymbol{x}_0) + \frac{1}{2} \sum_{i=1}^{m} (\boldsymbol{x}_i - \boldsymbol{x}_0)^T H_f(\boldsymbol{x}_0)(\boldsymbol{x}_i - \boldsymbol{x}_0)$$
$$+ o(\max_{i \in \{1,2,\ldots,m\}} \|\boldsymbol{x}_i - \boldsymbol{x}_0\|^3)$$
$$= m f(\boldsymbol{x}_0) + \nabla f(\boldsymbol{x}_0)^T (m\boldsymbol{x}_0 - m\boldsymbol{x}_0) + \frac{1}{2} \sum_{i=1}^{m} (\boldsymbol{x}_i - \boldsymbol{x}_0)^T H_f(\boldsymbol{x}_0)(\boldsymbol{x}_i - \boldsymbol{x}_0)$$
$$+ o(\max_{i \in \{1,2,\ldots,m\}} \|\boldsymbol{x}_i - \boldsymbol{x}_0\|^3)$$
$$= m f(\boldsymbol{x}_0) + \frac{1}{2} \sum_{i=1}^{m} (\boldsymbol{x}_i - \boldsymbol{x}_0)^T H_f(\boldsymbol{x}_0)(\boldsymbol{x}_i - \boldsymbol{x}_0) + o(\max_{i \in \{1,2,\ldots,m\}} \|\boldsymbol{x}_i - \boldsymbol{x}_0\|^3) \quad (8)$$
$$= m f(\boldsymbol{x}_0) + \frac{1}{2} \sum_{i=1}^{m} (\boldsymbol{x}_i - \boldsymbol{x}_0)^T H_f(\boldsymbol{x}_0) \boldsymbol{x}_i$$
$$- \frac{1}{2} \sum_{i=1}^{m} (\boldsymbol{x}_i - \boldsymbol{x}_0)^T H_f(\boldsymbol{x}_0) \boldsymbol{x}_0 + o(\max_{i \in \{1,2,\ldots,m\}} \|\boldsymbol{x}_i - \boldsymbol{x}_0\|^3)$$
$$= m f(\boldsymbol{x}_0) + \frac{1}{2} \sum_{i=1}^{m} (\boldsymbol{x}_i - \boldsymbol{x}_0)^T H_f(\boldsymbol{x}_0) \boldsymbol{x}_i$$
$$+ \frac{1}{2} (m\boldsymbol{x}_0 - \sum_{i=1}^{m} \boldsymbol{x}_i)^T H_f(\boldsymbol{x}_0) \boldsymbol{x}_0 + o(\max_{i \in \{1,2,\ldots,m\}} \|\boldsymbol{x}_i - \boldsymbol{x}_0\|^3)$$
$$= m f(\boldsymbol{x}_0) + \frac{1}{2} \sum_{i=1}^{m} (\boldsymbol{x}_i - \boldsymbol{x}_0)^T H_f(\boldsymbol{x}_0) \boldsymbol{x}_i + o(\max_{i \in \{1,2,\ldots,m\}} \|\boldsymbol{x}_i - \boldsymbol{x}_0\|^3)$$

□

### A.3 GENERALIZABILITY

In our research, we primarily examine the model's generalizability in two aspects: one is the generalization over the distribution of box dimensions (length, width, height), and the other is the generalization over the amount of boxes.

Table 6 presents the results of the model's generalizability over box dimensions, with the base model parameters set as $L = 120, W = 100, N = 100$. The first row labeled $a, b$ denotes that the lengths, widths, and heights of the boxes in the test dataset are generated from a uniform distribution of integers ranging from $a$ to $b$. We observe that, akin to other neural networks, the neural policy exhibits a certain degree of generalizability within a particular range. Despite the absence of boxes with dimensions of 5-9 and 51-55 in the training data, the model still performs remarkably well. However, when the divergence between the training data distribution and the testing data distribution becomes significant, performance degrades for both the base model and our proposed model.

Table 7 illustrates the model's generalizability with respect to the number of boxes, with the base model parameters set as $L = 120, W = 100, N = 100$. The first row marked $N_t$ refers to the number of boxes in the test set. We find that when $N_t = 200$, the model's generalizability is relatively commendable. Nevertheless, when $N_t = 400$ or $N_t = 800$, the model's performance tends to deteriorate.

Table 6: The experimental outcomes for varying $m$ with $L = 120, W = 100, N = 100$. Both figures outside and inside parentheses exclude the % symbol. The figure outside the parentheses denotes the average loading rate following seven experimental repetitions, while the figure within the parentheses signifies the standard deviation.

| $a, b$ | 5,55 | 3,60 | 1,70 |
|---|---|---|---|
| A | 73.40(1.55) | 71.09(2.44) | 68.02(2.54) |
| A+P+T | **76.37**(1.02) | 73.35(2.36) | 69.27(2.93) |
| A+LP+T | 76.08(3.70) | **74.08**(1.85) | **69.39**(3.17) |

Table 7: The experimental outcomes for varying $m$ with $L = 120, W = 100, N = 100$. Both figures outside and inside parentheses exclude the % symbol. The figure outside the parentheses denotes the average loading rate following seven experimental repetitions, while the figure within the parentheses signifies the standard deviation.

| $N_t$ | 200 | 400 | 800 |
|---|---|---|---|
| A | 72.84(2.16) | 70.78(2.37) | 64.97(5.09) |
| A+P+T | 75.60(3.83) | 71.58(2.06) | 66.30(3.79) |
| A+LP+T | 75.10(2.39) | 71.37(2.00) | 65.40(4.68) |

