# OpenReview forum: "Permutations improve performance in three-dimensional bin packing problem"
_ICLR.cc/2024/Conference — ICLR 2024 Conference Withdrawn Submission_

### Official Review · Reviewer_Pj8c · 2023-10-31

**Soundness:** 4 excellent
**Presentation:** 2 fair
**Contribution:** 3 good
**Rating:** 5
**Confidence:** 3

**Summary:**

This work presents a novel deep learning algorithm designed to tackle the challenging online 3D bin packing problem (BPP), a crucial combinatorial optimization task with applications across various industries. The inherent complexity of this problem stems from its vast combinatorial search space, which can lead to performance limitations in traditional operations research-based approaches, particularly in terms of speed and the need for domain-specific expertise.

Recent advancements in neural combinatorial optimization offer a promising avenue to address these limitations. Notably, deep reinforcement learning (DRL) methods can learn policies as effective combinatorial solvers by iteratively interacting with the problem environment, eliminating the reliance on problem-specific domain knowledge. However, it's worth noting that existing DRL techniques for combinatorial optimization have predominantly concentrated on vehicle routing problems (VRPs), leaving the 3D BPP relatively unexplored. Nonetheless, solving the 3D BPP still demands the development of specific techniques.

In this work, we introduce a permutation packing method and provide a mathematical proof demonstrating its capacity to enhance the approximation capabilities of deep neural networks. Given the potentially vast number of permutations involved, which could lead to increased memory consumption, we also propose a memory-efficient approach for permutation packing. Our experimental results empirically validate the effectiveness of these ideas, underscoring their practical relevance.

**Strengths:**

1. The idea of permutation packing is simple but novel.

2. Writing is clear and enjoyable to read.

3. Theoretical analysis is intuitive and highlights the motivation of the algorithm.

4. Ablations are deeply done and highlight the effectiveness of the method.

**Weaknesses:**

1. Algorithm pseudo code is too messy. It's really hard to read. Please improve the readability of the algorithm.

2. Empirical results are not sufficient. Please do more experiments in different scenarios (e.g., distributional shift, scale shift).

3. Real-world bin packing benchmark results are needed to verify performances.

4. This work is focused on the bin packing problem. The author should care about the extension of this paper's findings into a similar domain.

**Questions:**

1. Can this algorithm scale up to a larger scale? (e.g., $N>1000).

2. Can neural policy make good generalization capability to the unseen distribution of scales?

3. The reinforcement learning baseline is the rollout baseline. Did you try a more recent baseline for neural combinatorial optimization, such as a shared baseline [1,2]?


[1] Kwon, Yeong-Dae, et al. "Pomo: Policy optimization with multiple optima for reinforcement learning." Advances in Neural Information Processing Systems 33 (2020): 21188-21198.

[2] Kim, Minsu, Junyoung Park, and Jinkyoo Park. "Sym-nco: Leveraging symmetricity for neural combinatorial optimization." Advances in Neural Information Processing Systems 35 (2022): 1936-1949.

---

> ### Author Response · Authors · 2023-11-19
>
> Thank you for acknowledging our work. We have uploaded a revised version in response to your feedback.
>
> **Algorithm pseudo code being too messy:**
>
> In the revised version, we have removed the rollout algorithm section from the pseudo code for clarity.
>
> **Insufficiency of empirical results:**
>
> In the revised version, we have supplemented our experiments with distributional shift and scale shift in Section A.3.
>
> **Questions 1:**
>
> In Section 6 of the revised version, we discuss how our method can be extended to two-dimensional packing problems.
>
> **Questions 2:**
>
> Indeed, RCQL can handle large-scale packing. However, RCQL employs heuristic rules, randomly selecting 10-30 boxes each time and then using deep reinforcement learning to choose one among these 10-30 boxes. We are not sure if this meets your requirements.
>
> **Regarding the generalization capability of the neural policy to unseen distribution scales:**
>
> The neural policy, like most neural networks, can generalize well if the unseen distribution is close to the distribution of the training set. If the unseen distribution is far from the training set distribution, the performance may not be ideal.
>
> **Questions 3:**
>
> |P|LP|POMO|SYM|P+POMO|LP+POMO|P+SYM|LP+SYM|
> |---|---|---|---|---|---|---|---|
> |76.76|76.39|76.20|76.20|77.57|77.90|77.06|77.69|
>
> From the experimental results, our method is comparable to POMO and SYM-NCO. Moreover, our method can be integrated into POMO and SYM-NCO.

---

> ### Comment · Reviewer_Pj8c · 2023-11-20
> **Your rebuttal is commendable, but there is still some room for improvement.**
>
> I appreciate your efforts in providing the rebuttal and conducting additional experiments. The revised paper has shown significant improvements compared to the original submission.
>
> While the paper exhibits promise, it is not yet in its final, polished form to secure acceptance in this particular venue.
>
> Additionally, it may be beneficial for the author to consider incorporating other, similar tasks to further validate the proposed idea.

---

### Official Review · Reviewer_3QSB · 2023-10-31

**Soundness:** 2 fair
**Presentation:** 1 poor
**Contribution:** 1 poor
**Rating:** 3
**Confidence:** 5

**Summary:**

This paper aims to address the 3D Bin Packing Problem, where the output of an good neural network should remain unchanged when the length, width, and height of the boxes are interchanged. Based on this premise, the authors propose an arrangement packing method to enhance the testing capability of deep reinforcement learning methods.

**Strengths:**

1. The authors introduce additional perturbations, conducting multiple searches for solutions to the offline bin packing problem. This approach, building upon the existing baseline's ability to output feasible solutions in a single run, enhances the performance of the original packing method.

**Weaknesses:**

1. The primary idea of the authors, leveraging the symmetry of boxes to conduct multiple searches for solutions, results in excessive packaging and applies too much complex theory. The authors should simplify their presentation.
2. Conducting multiple searches for solutions and comparing them with the baseline increases both time consumption and requires more data utilization. This type of performance comparison is not fair. Additionally, the reviewers noticed that the authors did not report the time taken for various methods in the article, which is misleading.
3. In the supplementary materials, the authors only submitted their own method and did not provide the baseline algorithm. This is really a concern for the reviewer. The reviewer hopes that during the revision, the authors will submit the baseline code, and this will be considered for a potential score increase.

**Questions:**

The reviewer has no further queries.

---

> ### Author Response · Authors · 2023-11-19
>
> Thank you for reviewing our paper. We have uploaded a revised version.
>
> **Multiple Searches:**
>
> Both Permutation Packing and Limited-Memory Permutation Packing only use permutations to rectify the policy function, still only sampling one solution each time, rather than sampling multiple solutions from the policy function.
>
> **Report the Time:**
>
> In the revised version, Table 5 reports the time taken by various methods.
>
> **baseline code:**
>
> We have uploaded the baseline code for RCQL and Attend2pack.

---

> > ### Comment · Reviewer_3QSB · 2023-11-20
> > **Concern for reproducibility**
> >
> > The reviewer appreciates the authors' prompt response. However, in the modified baseline code, the reviewer still hasn't found the complete code for various baselines such as GA+DBLF, EP, LAFF, EB-AFIT, MTSL, MM, etc. As code reproducibility is the primary concern, the reviewer hopes to see these codes.

---

### Official Review · Reviewer_PXmX · 2023-10-31

**Soundness:** 3 good
**Presentation:** 1 poor
**Contribution:** 3 good
**Rating:** 5
**Confidence:** 2

**Summary:**

The paper proposes two methods, Permutation Packing and Limited-Memory Permutation Packing, to enhance the performance of solving the three-dimensional bin packing problem using deep reinforcement learning. These methods utilize permutations to approximate policy functions more effectively and can be integrated with existing models. The effectiveness of these methods is demonstrated through theoretical and experimental analysis, showing improved performance without the need for retraining the model.

**Strengths:**

1. The paper tries to address an important problem in the field of optimization and combinatorial optimization, namely the three-dimensional bin packing problem. The paper proposes a novel method for solving the three-dimensional bin packing problem using permutations to better approximate policy functions. This approach is relatively new and innovative, as most current works in the field involve modifications to the network architecture or the incorporation of heuristic rules.

2. The methods are motivated by theoretical analysis, which is supported by the experiments including sufficient baselines.

**Weaknesses:**

1. This paper appears to have been hastily prepared and contains numerous typos. E.g., "." on top of Sec. 3.2; Missing "." in the caption of Figure 1.

2. It lacks discussion on the limitations.

3. The bounded function assumption is kind of strict.

**Questions:**

1. What are the limitations?

2. Can the bounded function assumption be relaxed?

---

> ### Author Response · Authors · 2023-11-19
>
> We appreciate your review of our paper and your recognition of our work. A revised version has been uploaded.
>
> **numerous typos**
>
> In the revised version, we have made every effort to correct such grammatical and punctuation errors.
>
> **There is a lack of discussion regarding its limitations.**
>
> In the revised version, Table 4 reveals that one limitation of Permutation Packing is its higher memory demand, while Table 5 shows that Limited-Memory Permutation Packing increases the runtime by approximately 7%.
>
> **The assumption of a bounded function is somewhat strict.**
>
> For combinatorial optimization problems, the assumption of a bounded function is not strict as the output of the neural network is probability, which falls within the 0-1 range. However, it is indeed possible to remove the constraint of a bounded function.

---

### Official Review · Reviewer_5pxn · 2023-11-01

**Soundness:** 2 fair
**Presentation:** 2 fair
**Contribution:** 2 fair
**Rating:** 3
**Confidence:** 3

**Summary:**

The paper proposes Permutation Packing, as an decorator of the policy network in deep reinforcement learning, to introduce a permutation invariant mechanism for 3D bin packing problem. In particular, Permutation packing utilizes the symmetry and generate the policy using permutated replicas of the state representation. To reduce the memory overhead, the paper further proposes using Taylor's series to approximate the replica results. In experiments, the paper shows Permutation Packing is able to improve the performance of existing deep reinforcement learning methods for 3D bin packing in both training and test.

**Strengths:**

* The paper tries to use symmetry, which has been the core in modern science, to improve the policy in deep reinforcement learning for 3D bin packing.
* The experiment results show that the proposed method can not only improve the training process, but also works as a plug-in to improve policy quality in inference.

**Weaknesses:**

* The presentation of the paper is poor.
    *  Missing details in methods. For example, the paper does not have formal definition of the state representation. After reading the paper, I am still not clear how the permutation are performed on states. Also, using the hessian matrix of a deep neural network could be difficult, the paper should elaborate more on limited-memory permutation.
    * Missing details in experiments. The paper only have one experiments, which seems follows the experiment in Attend2Pack in section 3.5. However, the paper does not provide any details like how the boxes are generated and how the training and evaluation are performed. Also, the performance of the baselines are different from the number reported in the original papers, making the experiment results less convincing.
    * The paper spends a lot of space on materials with low relevance. For example, the approximation theory in section 2 and section 3 is necessary to appear in the main text. Using one sentence to introduce the norm $||\cdot||_{\Omega, p}$ and the remaining can be put into appendix.

* The soundness of the paper need more justification. The machine learning community has devoted significant effort for the symmetry in neural network. Some results, for example, using neural networks with special architectures like deep sets [1], self-attention [2], DSS layers [3]. For me, using an architected neural network admits symmetry is more straightforward than using the method proposed in this work. The paper should add more discussion, or experiments if applicable, about the difference and the advantage of Permutation Packing compared to these methods.


>[1] Zaheer, Manzil, et al. "Deep sets." Advances in neural information processing systems 30 (2017).\
[2] Vaswani, Ashish, et al. "Attention is all you need." Advances in neural information processing systems 30 (2017).\
[3] Maron, Haggai, et al. "On learning sets of symmetric elements." International conference on machine learning. PMLR, 2020.

**Questions:**

* What is the state representation and how is the permutation performed?

---

> ### Author Response · Authors · 2023-11-19
>
> We are grateful for your assessment of our paper, and we have uploaded a revised version in response to your feedback.
>
> **Questions:**
>
> The state S encapsulates both the length, width, and height of all boxes $X \in R^{n \times 3}$, where $n$ is the number of boxes and $3$ represents the dimensions of each box, as well as the coordinates of the boxes within the larger bin. Since the permutation doesn't affect the latter, we simplify the input to the neural network as $X$, denoting the output of the neural network as $f(X) \in R^n$.
>
> There are two types of permutation symmetries in bin packing problems:
>
> 1, Symmetry in the order of the input boxes: Each component of $f(X)$ corresponds to the probability of selecting the box. When a permutation is applied to the order of the input boxes, the output of the neural network should equal the original output subjected to the same permutation.
>
> 2, Symmetry in the dimensions of the boxes: We expect the output to remain the same when the length, width, and height of each box are interchanged. As there are six permutations for each box, this symmetry corresponds to a total of $6^n$ invariant permutations.
>
> We've already addressed the first symmetry with self-attention. Our method aims to enhance the second symmetry. We only interchange the dimensions of the boxes, not their order. Using deep sets, self-attention, and DSS layers cannot simultaneously satisfy both symmetries.
>
> Hessian Matrix:
>
> We don't need to directly compute the Hessian matrix, as $(x_i-x_0)^T H_f(x_0) \ x_i = (x_i-x_0)^T \nabla (\nabla f(x_0)^T \ x_i)$ can be computed by taking the gradient twice. In the revised version, we explain using eq (4) how we can avoid directly calculating the Hessian matrix.
>
> Missing details in experiments:
>
> The additional dataset used in Attend2Pack consists of boxes that perfectly form a rectangular prism, with the prism's length and width equal to that of the larger box. Such datasets are rarely seen in real-world scenarios.
>
> In the original paper, our experimental details are in A.4, which includes how we generate our test data and other training hyperparameters. Training data and test data are generated with equal probability from integers between 10 and 50; we conducted more than one experiment, and an additional experiment is in A.3 of the original paper. In the revised version, we move the experimental details to Sec 5.1, and the other experiment to Sec 5.2.
>
> The reasons our experimental results differ from the original Attend2Pack are: 1) the original paper did not release source code and evaluation datasets. 2) In the original paper, the box dimensions range from 20 to 80, which is larger than ours, and the larger box's dimensions are 100, which is smaller than ours. Larger boxes and smaller containers reduce the action space, making it easier to converge.
>
> Approximation Theory:
>
> In line with your suggestion, we have relocated most of the Approximation Theory content to the Appendix, and only included the evaluation metrics from the Approximation Theory in the main text.